# Evaluation of Anesthetic Specific EEG Dynamics during State Transitions between Loss and Return of Responsiveness

**DOI:** 10.3390/brainsci12010037

**Published:** 2021-12-28

**Authors:** Matthias Kreuzer, Tobias Kiel, Leonie Ernst, Marlene Lipp, Gerhard Schneider, Stefanie Pilge

**Affiliations:** Department of Anesthesiology and Intensive Care, School of Medicine, Klinikum rechts der Isar, Technical University of Munich, 81675 Munich, Germany; m.kreuzer@tum.de (M.K.); Tobias_Kiel@gmx.de (T.K.); Leonie.Ernst@mri.tum.de (L.E.); marlene.lipp@tum.de (M.L.); gerhard.schneider@tum.de (G.S.)

**Keywords:** anesthesia, anesthesia emergence, electroencephalogram, monitoring

## Abstract

Purpose: electroencephalographic (EEG) information is used to monitor the level of cortical depression of a patient undergoing surgical intervention under general anesthesia. The dynamic state transitions into and out of anesthetic-induced loss and return of responsiveness (LOR, ROR) present a possibility to evaluate the dynamics of the EEG induced by different substances. We evaluated changes in the EEG power spectrum during anesthesia emergence for three different anesthetic regimens. We also assessed the possible impact of these changes on processed EEG parameters such as the permutation entropy (PeEn) and the cerebral state index (CSI). Methods: we analyzed the EEG from 45 patients, equally assigned to three groups. All patients were induced with propofol and the groups differed by the maintenance anesthetic regimen, i.e., sevoflurane, isoflurane, or propofol. We evaluated the EEG and parameter dynamics during LOR and ROR. For the emergence period, we focused on possible differences in the EEG dynamics in the different groups. Results: depending on the substance, the EEG emergence patterns showed significant differences that led to a substance-specific early activation of higher frequencies as indicated by the “wake” CSI values that occurred minutes before ROR in the inhalational anesthetic groups. Conclusion: our results highlight substance-specific differences in the emergence from anesthesia that can influence the EEG-based monitoring that probably have to be considered in order to improve neuromonitoring during general anesthesia.

## 1. Introduction

Patient monitoring based on the electroencephalogram (EEG) provides an easy-to-apply approach in a clinical setting that may help to adjust anesthetic drugs to specific individual requirements for general anesthesia, i.e., to avoid too light or excessively deep anesthesia. Commercial devices, such as Bispectral Index (Medtronic, Dublin, Ireland) [1], SEDLine (Masimo, Irvine, CA, USA) [2], Entropy Module (GE Healthcare, Helsinki, Finland) [3], or Cerebral State Index (CSI) of the CSM monitor (Danmeter ApS, Odense, Denmark) [4] can provide a dimensionless index (e.g., from 0 to 100) that inversely correlates with the patients’ level of cortical depression. It has to be mentioned that these indices rely on highly processed EEG information and hence it is not clear how they react to EEG changes right at a state transition or to substance specific differences. Inadequate anesthetic titration may severely influence patient outcome—too light anesthetic levels can lead to unintended wakefulness and too deep levels, especially with burst suppression EEG, may be associated with postoperative delirium (POD) [5,6]. Thus, the European Society of Anaesthesiology (ESA) evidence-based and consensus-based guidelines on POD recommend EEG based monitoring of anesthetic depth in all patients to avoid too deep general anesthesia, i.e., burst suppression [7]. While the raw EEG characteristics or the spectral representation of the EEG over time allows to understand the distinct, anesthetic-induced changes of this signal, the indices may not be able to fully reflect them. Further, known time delays in index calculation [8,9] can lead to unreliable indices during dynamic phases, such as anesthesia induction and emergence. Another issue are differences in EEG features that may also impact the one-fits-all indices. Different anesthetic substances induce different EEG patterns [10,11,12]. Demographic factors, such as age, alter the EEG characteristics and influence the indices [13,14,15,16,17]. When waking up from anesthesia the changes in EEG activity throughout emergence can follow different trajectories [5,18,19]. Because the type of trajectory seems to correlate with the risk of postoperative pain [18] or delirium in the postoperative care unit [5], monitoring the emergence phase is of special interest. Here, we present results from the analysis of a data set [20] that can help to understand the impact of anesthetic substance on processed EEG parameters, like spectral features, permutation entropy (PeEn) [21], and CSI [4] as a proxy for commercial monitoring applications. 

## 2. Materials and Methods

### 2.1. Study Design

With our approach, we tried to analyze EEG behavior during the transitions between loss (LOR) and return of responsiveness (ROR). We chose to analyze the original data of a previous patient study, which is particularly suited to provide the key data (i.e., different anesthetics for maintenance of general anesthesia and accurate clinical verification of consciousness). We initially designed the study to evaluate the performance of the CSI during LOR and ROR. The Ethics Committee of the Technical University of Munich, Munich, Germany (Chairman Prof. A. Schömig) approved the study (Ethical Committee N° 1239/05) and we initiated the study on 2 March 2005. We included 45 adult patients with an American Society of Anesthesiologists physical status I or II, undergoing elective surgery under general anesthesia. All patients gave informed written consent to the study. 

We defined the following exclusion criteria: history of neurological or psychiatric disease, medication known to affect the central nervous system including drug or alcohol abuse, and the indication of a rapid sequence induction (e.g., pregnancy, emergency). According to standard clinical practice, each patient received one of three pre-defined combinations of anesthetic drugs, our three study groups: sevoflurane group (sevoflurane/sufentanil), isoflurane group (isoflurane/sufentanil), and propofol group (propofol/remifentanil) with *n* = 15 patients each. The attending anesthetist chose the anesthetic, relying on standard operating procedures for the particular surgical types, and in consideration of preexisting conditions. We used propofol for induction in all patients. We administered atracurium or mivacurium as neuromuscular blocking agent. There was no randomization in order to reflect standard clinical practice. 

### 2.2. Monitoring

We used a Datex^®^ AS/3 (GE Healthcare, Chalfont St Giles, United Kingdom) compact monitor for non-invasive measurement of blood pressure, heart rate, oxygen saturation, inspiratory oxygen, end-tidal carbon dioxide, and volatile anesthetic concentrations and recorded demographic data and type of anesthetic premedication for each patient. For EEG recording and evaluation of index performance, we used a CSI monitor.

### 2.3. Clinical Protocol

Some patients received benzodiazepines for premedication according to clinical standard. The clinician in charge slowly induced anesthesia by intravenous injections of 20 mg propofol every 30 s. In order to define LOR and ROR, we asked the patient every 15 s to squeeze the investigator’s hand during anesthesia induction and emergence. In case of a positive response, we verified the response by an immediate repetition of the command. We defined LOR as the first missing response after a repeated command during induction and ROR as the first positive response during emergence. In case the patient showed a reproducible response, i.e., opening of the eyes or moving of the head, but not reaction to the “squeeze hand” command, this time was noted ROR. To ensure a preserved ability to react to the command in case of administered neuromuscular blocking agents, we applied Tunstall’s isolated forearm technique [22]. We delivered propofol, sevoflurane, or isoflurane for anesthesia maintenance. 

Clinical endpoints were a constant end-tidal volatile anesthetic concentration and respectively a constant propofol effect-side concentration determined with the pharmacokinetic model by Schnider et al. [23].

### 2.4. EEG Recording and CSI Calculation

For our analyses, we stored the CSI trend data as well as the raw EEG as provided by the CSM monitor. The stored raw EEG had a sample rate of 100 Hz and a frequency range from 6–42 Hz. We chose electrode positions as recommended by the manufacturer: at the middle of the forehead, at the left forehead and at the mastoid region on the left side, all with impedances of 5 kΩ or less (https://www.danmeter.dk/en/files/CSM-Monitor-MKII---Manual-561105003--US-only-.pdf accessed on 25 November 2021). We also stored the CSI, the burst suppression ratio (BS%), and the signal quality data on a laptop computer equipped with specific software (CSM-Capture^®^ 2.02, Danmeter, Odense, Denmark) using CSM Link™. The CSI is derived by a proprietary algorithm that utilizes an adaptive neuro-fuzzy inference system (ANFIS) to combine the information from the three spectral EEG parameters α-ratio, β-ratio, and β-ratio/α-ratio as well as BS% [4].

We recorded EEG and all related data synchronized to standard monitoring parameters and manually documented singular events like LOR/ROR in NeuMonD [24].

### 2.5. CSI and EEG Analysis

For our analyses that mainly focused on the EEG (and CSI) features during substance-specific anesthesia emergence, we calculated the power spectral density (PSD) for 10 s EEG episodes with a 1 s shift to construct density spectral arrays (DSA) and to derive the trends of power in the different EEG frequency bands. We further calculated the PeEn [21], a time domain measure that can be correlated to the power spectrum [25] of the recorded EEG. PeEn is one of the most powerful 1-channel EEG parameters to differentiate between consciousness and unconsciousness [26,27]. We calculated PeEn with embedding dimension m = 3 and time lag τ = 1 after a low pass filtering of the EEG to 30 Hz for 10 s EEG episodes with a 1 s shift as well. Because of the narrow EEG-bandwidth provided by the CSM, we confine our reporting to relevant findings in the 6–30 Hz range. We chose these settings for m and τ for PeEn calculation because of the direct link to the power spectrum [25].

We evaluated the performance of the parameters over the induction and throughout the emergence phase as well as the performance of PeEn and CSI to distinguish between a responsive and an unresponsive patient right at the state transitions LOR and ROR. Therefore, we compared the parameter values derived 15 s before the LOR/ROR event to the values 30 s after the event. The interval of 15 s to ask patients to squeeze hand allows a clear definition of LOR/ROR, and thus the conclusion that 15 s before LOR (ROR) the patient was responsive (unresponsive). We selected the index values +30 s after the event to account for the time delay of index calculation [8], i.e., to allow the EEG monitor to indicate the current clinical state. This approach is consistent with previously published data [28,29]. It has to be mentioned that the derived results between CSI and PeEn are not comparable because of a varying and considerable time delay in index calculation of the CSI [8,9].

### 2.6. Statistical Analysis

To evaluate spectral differences, we used the area under the receiver operating curve (AUC) together with 10k-fold bootstrapped 95% confidence intervals. In case the 95% confidence interval did not contain 0.5 (i.e., ‘no effect’), the comparison was defined to be significantly different. Hentschke and Stuettgen also showed this relationship [30]. To avoid the discussion of false positives, we only considered findings significant if, in case of comparing PSD, at least two neighboring frequency bins showed significant differences. This or a similar approach has been used before [10,14]. For comparison of the density spectral arrays, we only discuss differences if they occurred in clusters, because of the low probability of a concentrated occurrence of spurious false positives in a cluster.

We used the Wilcoxon signed rank test (significance level: *p* < 0.05) and the effect size Hedges’ g for dependent data to test for differences in CSI and PeEn before and after LOR/ROR. We also used the AUC and 95% confidence intervals to evaluate the parameters’ (PeEn and CSI) performance to separate between responsiveness and unresponsiveness, because the AUC for dichotomous data is equivalent to the prediction probability Pk, commonly used to assess the performance of neuromonitoring approaches [31]. For analysis of the demographic data, the duration of CSI ≥ 80 before ROR, and the differences in EEG-band power during anesthesia emergence and ROR, we used a Kruskal–Wallis test with Dunn’s post-hoc test to correct for multiple comparisons [32]. For comparison of sex and ASA distribution between groups, we used a Freeman–Halton test. We performed all statistical tests with MATLAB also using the MES toolbox [30].

## 3. Results

### 3.1. Demographics

There were no significant differences between the groups in age, size, gender, and ASA status. Patients in the isoflurane group had a significantly higher weight than patients in the sevoflurane (*p* = 0.036) and propofol (*p* = 0.002) group, but there was no significant difference in BMI among the groups. Table 1 contains the detailed information regarding the demographics. In the isoflurane group, 12 patients received dormicum, (+ additional tranxilium in six cases and one patient received extra nexium) and in the sevoflurane and propofol group, 11 patients each received dormicum (+ two received additional tranxilium and one extra nexium) for premedication. The duration from the begin of emergence period to ROR was 20 min 36 s (11 min 14 s–49 min 48 s) for isoflurane, 16 min 29 sec (9 min 55 s–39 min 05 s) for sevoflurane and 14 min 28 sec (7 min 0 s–27 min 30) for propofol.

### 3.2. EEG and Index Features during Propofol-Induced Loss of Responsiveness

The EEG during the propofol induced LOR showed the characteristic slowing of the EEG and the onset of strong oscillatory activity in the EEG alpha-band. Figure 1A shows the median DSA during LOR for all patients. When looking at certain time points before and after LOR, i.e., when comparing the power spectral density (PSD) extracted 15 s before LOR to the PSD 30 s after the event, we could observe significantly higher alpha power and lower power in the beta band frequencies after LOR (Figure 1B). These changes in the EEG impacted the processed EEG parameters PeEn and CSI, which significantly and strongly decreased during LOR (PeEn: *n* = 45; *p* < 0.001, g = 0.94 (0.65 1.8); CSI: *n* = 43; *p* < 0.001, g = 1.13 (0.85 1.519) (Figure 1C).

We also compared the spectral parameters between the patients in the different maintenance anesthetic regimen groups to ensure comparable conditions. We did not find significant differences between the groups as displayed in Appendix A. Table 2 contains the statistical details.

### 3.3. Substance Specific Differences in the EEG and CSI Features during ROR

We found differences in the EEG spectrum and the processed EEG parameters during anesthesia emergence towards ROR. Figure 2 displays the DSA from 10 min before until ROR for the three anesthetic groups as well as the statistical differences between the groups. At the start of emergence all groups show strong oscillatory activity in the EEG alpha-band, but in contrast to propofol, the patients receiving volatiles had higher power in the higher frequencies as well. This higher power persisted throughout emergence. While towards the end of emergence at around 2 min before ROR there were no more differences between the propofol and sevoflurane group, the differences between propofol and isoflurane remained (in parts until immediately before ROR). Between the isoflurane and sevoflurane group, we did not observe any significant differences between emergences (Figure 2). To complete the picture, we also assessed the differences in the normalized power and found the results to be weaker than for the absolute power. The corresponding plots can be found in Appendix A.

The difference in EEG changes towards ROR was also reflected in the behavior of the processed EEG parameters. For PeEn, we observed no significant change throughout the immediate transition to ROR. For isoflurane and propofol the PeEn did not increase significantly (isoflurane: *p* = 0.057 propofol: *p* = 0.058), but the effect size Hedges’ g indicated a *medium* effect without the 95% CI containing 0 (isoflurane: g = −0.52 (−1.22 −0.03); propofol: g = −0.68 (−1.32 −0.17)). There was no significant effect in the sevoflurane group (*p* = 0.216). The details are presented in Figure 3A–C. For the CSI, we did not find a significant change in the index from 15 s before until 30 s after ROR for the patients in the isoflurane (*p* = 0.234) and sevoflurane (*p* = 0.445) group. For propofol, we found a significant (*p* = 0.023) CSI increase of *medium* strength (g = −0.71 (−1.20 −0.33)). Figure 3D–F presents the corresponding connected dot and box plots. Table 2 presents the details. We also found significantly higher CSI and PeEn values (*p* < 0.001) after ROR than before LOR. The corresponding plots are presented in the supplements as Appendix A.

### 3.4. Implications of Substance-Specific Differences on Monitoring

The earlier activation of higher EEG frequencies during anesthesia emergence influenced the CSI, used as a proxy for the monitoring systems. The median duration the CSI reflected wakefulness, CSI ≥ 80, before ROR was 73 s [43 s 203 s] for isoflurane, 101.5 s [0 s 143.5 s] for sevoflurane, and 24 s [2 s 51 s] for propofol. There was no significant difference between the durations (*p* = 0.3145, Chi-square = 2.31). Three patients in the isoflurane, four patients in the sevoflurane group, and two patients in the propofol group had CSI < 80 at the time of ROR, included in the analysis as a duration of 0 s. After removing these patients from this analysis, the median duration was 190.5 s (64.8 s 289.3 s) for isoflurane, 143.5 s (101.5 s 222.5 s) for sevoflurane, and 25 s (4 s 79.5 s) for propofol. These durations were significantly different (*p* = 0.0145, Chi-square = 8.46) with a significant shorter duration of CSI ≥ 80 for propofol when compared to isoflurane (*p* < 0.001, AUC = 0.83 (0.55 0.98)). There was no significant difference between the sevoflurane and propofol group, but the AUC = 0.80 (0.57 97) revealed a *strong* effect of substance on CSI ≥ 80 before ROC. Figure 4 presents the according box plot.

## 4. Discussion

With our analyses, we could show that the EEG changes during anesthesia emergence and throughout the return to responsiveness depend on the maintenance anesthetic regimen. To investigate differences, we concentrated on the spectral changes in the EEG, from 10 min before ROR until ROR, as well as on the behavior of processed EEG indices, PeEn and CSI, immediately around ROR. In addition to the ROR transition for three anesthetic regimens, we also investigated the EEG changes during anesthesia induction with propofol. For the induction, we could observe slowing of EEG [33] and the development of strong oscillatory activity in the EEG alpha band, typical for propofol [10]. This change of the EEG towards slower oscillatory throughout LOR activity led to the significant decrease of PeEn and CSI. The LOR, in general, is a relatively fast process, triggered by administration of hypnotic agents (propofol) directly into the central compartment. Rapidly increasing effect-site concentrations affect critical structures in a way that leads to unconsciousness and that promptly modifies the EEG, facilitating relatively profound deviations in EEG-based parameters and indices as observed within the predefined time interval (e.g., LOR-15s to LOR+30s). We also used the LOR transition to check for the absence of significant differences in the EEG among the groups to ensure the differences described during anesthesia emergence were not due to predisposed differences. This helps us attribute our findings during anesthesia emergence to the substance and to heterogeneous samples. The ROR, in contrast to LOR, is an internal and slow process, primarily determined by pharmacokinetic properties of the maintenance anesthetics. Drug redistribution and elimination from tissues dependent on the cumulatively administered amount (propofol), or alveolar ventilation (volatiles) [34,35]. The state change during ROR back to consciousness may include the task of overcoming the orexinergic blockade and hence emergence may even be deemed an active process [36]. Hence, the substance-specific differences in the EEG towards ROR may be caused by the different modes of action [37,38] and the basic pharmacokinetic properties of the administered anesthetics [39]. Although both anesthetics seem to modulated the activity in the thalamocortical loop [37], the impact of propofol and sevoflurane on these structures may be different [38] as it may be for isoflurane as well [40]. The EEG features between these anesthetics are different as well under general anesthesia without [10,11] and with burst suppression [20,41]. The effect of propofol on the cortex may be less pronounced [38,42]. If volatiles affect the cortical activity in a stronger fashion, the changes during emergence may be stronger as well, possibly leading to the activation of higher frequencies. The EEG changed differently for the anesthetic regimen investigated with the inhalational anesthetics causing an activation of EEG frequencies in the EEG beta-band range earlier than propofol. At the beginning of the emergence period, the observed strong alpha band activity most probably reflected the described alpha anteriorization during general anesthesia and the hyperpolarized thalamocortical state which also is typical for general anesthesia [43]. The beta activation may point to an activation of cortical activity, but more research needs to be conducted to thoroughly identify possible mechanisms. Further, the “quality” of activation needs to be assessed in the future to be able to distinguish between states of increased activation or higher entropy and states of organized complexity. For instance, low doses of propofol may also lead to increases in (Lempel-Ziv) complexity when compared to an awake baseline [44]. Because the role of neural inertia seems to become more important when investigating state transitions, we need to better understand the mechanisms of the single state transitions [45]. The parameter values were significantly higher after LOR than before ROR. This has been shown for the CSI before [46] at similar time points. This difference may also be attributed to neural inertia.

### 4.1. Impact on EEG-Based Monitoring

Of course, the difference in the emergence EEG characteristics also influence parameters or indices used for monitoring. Here we investigated the impact on PeEn, an entropic measure in the time domain that showed high performance in the separation between consciousness and unconsciousness [26] and the CSI, an index derived from spectral EEG parameters that is still used for patient monitoring. Although, there is no new version of the CSI available, the monitoring performance of this index is comparable to other devices [47,48]. For PeEn calculation, the upper limit of the EEG frequency range was 30 Hz to reduce the influence of possible EMG contamination [49]. We would like to state that monitoring the depth of anesthesia and tracking the state transitions might present different challenges to the monitors. To display the level of anesthesia a smooth and stable index may seem preferable, but it seems to come at the cost of a time delay of index calculation [9,50]. Therefore, the monitors may not be designed to reliably track the state transition or identify the time point of LOR or ROR. Still, the evaluation of the emergence behavior may be important in terms of outcome [5] and hence, the index reaction also seems of importance.

We observed similar results with PeEn and with CSI. Still, PeEn showed poorer performance during LOR. This may be due to the reduced frequency range of PeEn and the possible inclusion of EMG activity by the CSI. The CSM EMG index showed a very strong reaction on the LOR (Appendix A). In general, EMG seems to drive neuromonitoring devices to a substantial degree [51] and the EMG frequency range is not limited to frequencies above 30 Hz but can overlap with the EEG spectrum to a substantial degree [52]. Hence, our analyses may be influenced by motor activity, but the exact impact of EMG needs to be investigated using more sophisticated EMG montages to be able to separate EEG from EMG by e.g., independent component analysis. During ROR, the parameters seemed to perform best for propofol. This was mainly due to the observation that for the inhalational anesthetics, the parameters indicated values reflecting wakefulness prior to ROR, which was most probably caused by the earlier activation of higher frequencies as indicated by the DSA and the CSI ≥ 80. Consecutively, by choosing the ROR-15s to ROR+30s window, the calculated parameters did not reflect the clinically verified state transition. It has to be stated that our choice of time points before and after the state transition lead to a very challenging data set because the EEG and CSI information were extracted close to the transition. However, at the same time this approach possibly most realistically reflects the challenges for the monitors and parameters in daily life. The selection of time points of the EEG or index values to be used for comparison can influence the separation performance as reflected by the AUC. For the CSI, for example, the prediction probability, which is equivalent to the AUC [31], was 0.88 for the comparison of steady-state consciousness versus unconsciousness during LOR [47], whereas it was 0.75 for the separation during state transitions [28]. One issue that has to be considered for our substance-specific investigations is the fact that the indices and entropic parameters do not incorporate the information regarding the substance used. Since propofol and volatile anesthetics lead to different intraoperative EEG patterns [10] the starting conditions at the beginning of anesthesia emergence could have been different. The possibility of intraoperative, substance-specific differences, i.e., if the level of cortical depression is comparable, has to be investigated in the future. Considering basic EEG behavior, and differing performance of CSI and PeEn during anesthesia induction, and emergence under prevalent study conditions, basic pharmacologic aspects have to be considered. Co-administered opioids affect clinical signs of anesthesia emergence [53]. Our study protocol (volatiles/sufentanil vs. propofol/remifentanil) did not eliminate the potentially distorting effect of different opioids on LOR/ROR detection. Thus, it seems appropriate to refer to *group-specific* instead of *substance-specific* CSI and PeEn performances. *Group-specific* differences in the performance to detect ROR may have different reasons. Different substances, i.e., propofol and volatile anesthetics lead to different EEG characteristics during general anesthesia [10]. The spectral EEG characteristics between sevoflurane and isoflurane seem comparable [54]. Further, our result of different durations from CSI indicating awake until the patients’ response to the *squeeze hand* command seems to indicate a substance-specific difference in the time from the evolution of faster EEG oscillations indicative of cortical activity and a coordinated response by the patient. Our results suggest that this episode of uncoordinated cortical activity maybe longer with isoflurane and sevoflurane. A third possibility, which has to be evaluated in the future, may be different emergence trajectories for different substances. Previous research described different spectral EEG trajectories during anesthesia emergence and their resemblance to sleep stages [19] as well as their association with adverse outcomes [18,55]. EEG filter settings of the original study, however, did not allow classifying these trajectories.

### 4.2. Clinical Implications

We could identify anesthesia-regimen-specific differences in the behavior of processed EEG parameters during anesthesia emergence that may be attributed to a difference in the activation of higher EEG frequencies. These differences could lead to the impression of awake patients considerably before the ROR in our patients receiving volatile anesthetics. In case of reacting to this “awake information” and trying to address and wake up the patient, the natural process of emergence may be disrupted. Since the quality of emergence may influence the postoperative cognitive outcome [5], this could be counterproductive. Hence, the anesthesiologist should not only rely on the processed EEG information as a marker for a patient that returns to consciousness.

### 4.3. Limitations

First, our EEG was recorded with the CSM, which has a high pass at ~6 Hz. Hence, we cannot describe any effects in the EEG delta and low-theta band. Especially, the behavior of delta-band activity should be investigated in the future, because it is an important feature of the EEG under general anesthesia [56]. These low frequency ranges are considerably affected by general anesthetics. However, the changes observed in the available frequency range underlines the need for monitoring approaches adjusted to the anesthetic protocol and patient characteristics. Further, we can only report changes in a frontal, single-channel EEG that may limit the amount of information regarding changes in global brain electrical activity. Especially during state transitions, the EEG may be contaminated by muscle activity. Hence, we limited the frequency range for PeEn analysis to 30 Hz. Further, the CSI is not as commonly used as for example the BIS or the SEDLine. Nevertheless, it is derived by sub-parameters that process spectral information similar to the other monitors. Our results presented are based on the investigations of different anesthetic regimens that also contain different analgesic drugs, i.e., remifentanil and sufentanil. Hence, the impact of this difference in the regimens has to be investigated in the future. Previous results may suggest that the different opioids may not influence the performance of the bispectral index at the loss of consciousness in a different fashion [56]. Because we present results of the processed EEG, we cannot reveal any mechanistic processes during LOR or ROR. Another factor that needs to be considered in future research is the role of blood pressure on the EEG during anesthesia emergence.

## 5. Conclusions

The EEG features during anesthesia emergence differ, depending on the anesthetic maintenance regimen. The EEG during emergence from propofol anesthesia with remifentanil takes a different course compared to sevoflurane or isoflurane with sufentanil. Patients that received these inhalational anesthetics showed an EEG activation of higher EEG frequencies way before their clinically verified return of responsiveness. These differences influence the EEG based monitoring as well. To more reliably track the patients’ emergence in the future, the monitoring approaches may have to be adjusted for the anesthetic regimen used.

## Figures and Tables

**Figure 1 brainsci-12-00037-f001:**
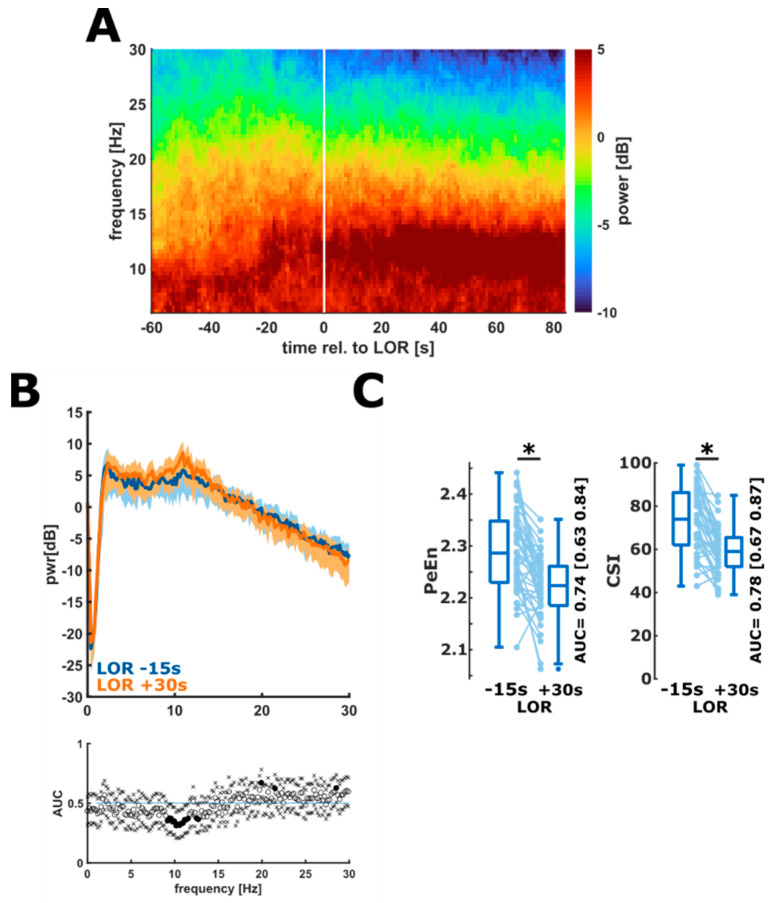
Changes in the spectral EEG representation and in processed EEG parameters during LOR. (**A**). Density spectral array (DSA) during the LOR episode. Over time, a strong alpha rhythm develops and the power in the faster frequencies decreases. (**B**) Comparison of the power spectral density 15 s before LOR (blue) to 30 s after LOR (orange). After LOR, the power in the alpha frequencies was significantly higher and there was power in the higher (beta) frequencies as indicated by the black dots in the AUC plot. (**C**) PeEn (left) significantly (*) and strongly decreased during LOR (*p* < 0.001, g = 0.94 (0.65 1.8) and the performance to separate responsiveness from unresponsiveness was “*fair*” (AUC = 0.74 (0.63 0.84)). CSI (right) significantly and strongly decreased during LOR (*p* < 0.001, g = 1.13 (0.85 1.51)) and the performance of the CSI to separate responsiveness from un responsiveness was “*fair*” (AUC = 0.78 (0.67 0.87)).

**Figure 2 brainsci-12-00037-f002:**
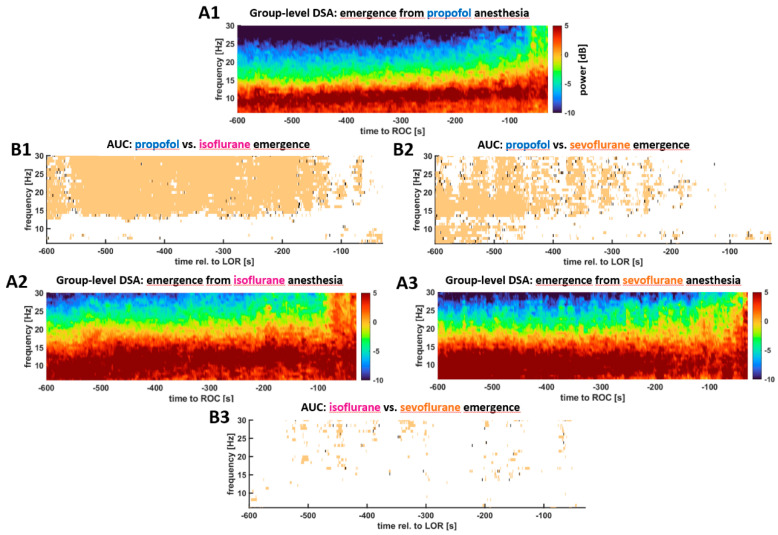
Density spectral arrays (DSA) plots for all three groups derived from the pooled (median) power spectral density of all patients within a group (**A**) and the corresponding information regarding a significant difference (**B**) as determined by the 95% confidence intervals of the AUC. **A**. DSA of anesthesia emergence for the different substance groups; **A1**. Group-level emergence DSA of the patients receiving propofol. **A2**. Group-level emergence DSA of the patients receiving isoflurane. **A3**. Group-level emergence DSA of the patients receiving sevoflurane. **B**. Maps of the pixel-wise AUC calculations. **B1**. Comparison between the propofol and isoflurane group. **B2**. Comparison between the propofol and sevoflurane group. **B3**. Comparison between the isoflurane and sevoflurane group. During emergence from propofol maintenance, the power in the frequencies above ~15 Hz was significantly lower than from maintenance with inhalational anesthetics (**B1**, **B2**). These differences persisted until very close to ROR when compared to the isoflurane group (**B2**). There were no significant differences in the emergence behavior between sevoflurane and isoflurane (**B3**).

**Figure 3 brainsci-12-00037-f003:**
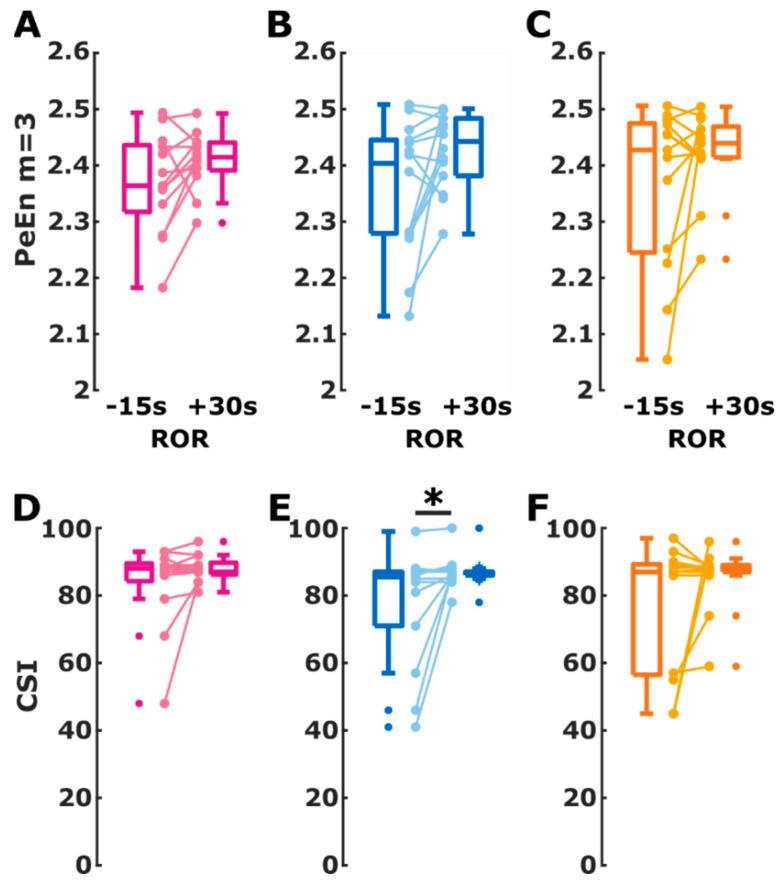
Change in cerebral state index (CSI) and permutation entropy (PeEn) values during return of responsiveness (ROR) from 15 s before ROR to 30 s after ROR for the isoflurane group (purple), propofol group (blue), and sevoflurane group (orange). The box plots indicate the distribution of CSI and PeEn values before and after ROR used for AUC calculation. The connected dots present the change in CSI and PeEn for each single patient. (**A**) The signed rank test indicated a non-significant change with a medium effect size of PeEn in patients assigned to the isoflurane group (*p* = 0.057; Hedge’s g = −0.52 (−1.22 −0.03)). (**B**) In the propofol group, the PeEn of the patients showed a non-significant (*p* = 0.0580) increase of medium effect (g = −0.68 (−1.32 −0.17)). (**C**) PeEn in the sevoflurane group increased non-significantly with a medium effect (*p* = 0.2163; g = −0.77 (−1 0.05)). (**D**) There was no significant different in the CSI change within the patients for the isoflurane group (*p* = 0.2344) (**E**) The CSI in patients in the propofol group significantly (*) increased (*p* = 0.0234; g = −0.71 (−1.20 −0.33), medium effect) (**F**) For the sevoflurane group the change within the patients was not significant (*p* = 0.4453).

**Figure 4 brainsci-12-00037-f004:**
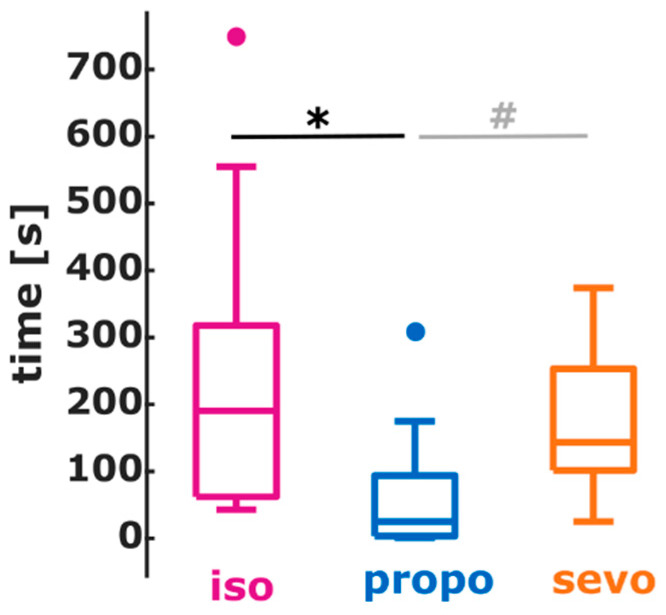
Boxplots representing the duration, the CSI indicated an awake patient before the return of responsiveness (ROR) for isoflurane (purple), sevoflurane (orange), and propofol (blue). For propofol, the duration was shortest with a significant (*, *p* < 0.001) difference compared to isoflurane and a (#, non-significant) but strong (AUC > 0.7) difference compared to sevoflurane.

**Table 1 brainsci-12-00037-t001:** Patient characteristics data of the three patient groups.

	Propofol Group *n* = 15	Sevoflurane Group *n* = 15	Isoflurane Group *n* = 15	*p*-Value
Age (y)	57 (45 67)	43 (38.5 63.5)	40 (33 46.5)	* *p* = 0.1008
Size (cm)	168 (161 178.5)	173 (167 179)	180 (173 185.5)	* *p* = 0.1059
Weight (kg)	71 (68 77)	75 (69.5 79.5)	90 (81 100)	* *p* = 0.0021
BMI	25.4 (23.2 31.2)	25.8 (23.8 36.6)	28.4 (25.8 34.1)	* *p* = 0.1474
LOC to ROC (min)	107 (89 115)	109 (93 182)	129 (103 142)	* *p* = 0.221
Sufentanil (µg)	n/a	25.0 (20.0 37.5)	30.0 (27.5 40.0)	† *p* = 0.391
Remifentanil (µg)	992 (867 1408)			
Gender (m/f)	7/8	9/6	11/4	^§^*p* = 0.3296
ASA (I/II)	10/5	8/7	10/5	^§^*p* = 0.6839

Data are presented as mean (first and third quartiles). * Kruskal–Wallis test; † Mann–Whitney U test; ^§^ Freeman–Halton test.

**Table 2 brainsci-12-00037-t002:** Statistical parameters for the cerebral state index (CSI) and permutation entropy (PeEn) analysis at loss and return of responsiveness (LOR/ROR).

	−15 s	+30 s	*p*-Value	Hedge’s g	AUC
LOR: CSI	74 [62 86]	59 [52 65]	*p* < 0.001	1.13 [0.85 1.51]	0.78 [0.67 0.87]
LOR: PeEn	2.29 [2.23 2.35]	2.22 [2.19 2.16]	*p* < 0.001	0.94 [0.65 1.8]	0.74 [0.63 0.84]
ROR-isoflurane: CSI	88 (48 to 93)	87 (81 to 96)	0.234	n.s.	0.50 [0.28 0.74]
ROR-sevoflurane: CSI	87 (49 to 97)	88 (59 to 96)	0.445	n.s.	0.59 [0.36 0.81]
ROR-propofol: CSI	85.5 (41 to 99)	87 (78 to 100)	0.023	−0.71 [−1.20 −0.33]	0.63 [0.17 0.58]
ROR-isoflurane: PeEn	2.36 [2.33 2.43]	2.41 [2.39 2.44]	0.057	0.52 [−1.22 −0.03]	0.63 [0.40 0.85]
ROR-sevoflurane: PeEn	2.43 [2.25 2.48]	2.43 [2.41 2.46]	0.216	n.s.	0.56 [0.32 0.78]
ROR-propofol: PeEn	2.40 [2.28 2.44]	2.44 [2.39 2.48]	0.058	−0.68 [−1.32 −0.17]	0.68 [0.44 0.85]

## Data Availability

Data can be requested from the authors.

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
