# Peer review of "Evaluation of Anesthetic Specific EEG Dynamics during State Transitions between Loss and Return of Responsiveness"

_brainsci, 2021, doi:10.3390/brainsci12010037_

Round 1

Reviewer 1 Report

Dear Editor

My comments to the article:

Manuscript ID: brainsci-1503037
Title: Evaluation of anesthetic specific EEG dynamics during state
transitions between loss and return of responsiveness

It is an interesting article that describes the behaviour of the frequencies of bands extracted from the EEG, with different gabaergic drugs and how, after analysing the phenomenon of unconsciousness (LOC), they study the anaesthetic awakening phase (ROC). The changes of the Cerebral State Index (CSI) monitor, and the Permutation Entropy (PeEn) are also evaluated.

A different awakening behaviour is demonstrated depending on the drug used due to the presence of significant beta bands in the case of inhaled drugs. These bands are not present in the case of Propofol.

The analysis is completed by describing the behaviour of CSI and PeEn during anaesthetic induction. Both show significant differences 15 seconds before LOC versus 30 seconds after LOC but during awakening the monitor can differentiate 15 seconds before clinical awakening (ROC) and 30 seconds after.

The author noted that a beta activity was present the last 10 min before ROC in the inhaled drugs (isoflurane, sevoflurane) but not in propofol.

The article is interesting to note the limited performance of global indices of brain activity analysis and assesses the use of band spectrography as more useful.

Unfortunately, the CSI is not a monitor that has had continuity and is not compared to newer versions such as CONOX.

The article is well designed and with adequate statistics for the available data. It is part of the development of studies that aim to find more specific markers for the LOC and ROC phenomena.

I think the article should be more explicit in its claim and limitations. Thus, giving a clearer picture of what indices, spectrography and future monitors should look for to better represent the clinical and informatics phenomenon, and to guide the reader in the interpretation of the results.

I also miss in the study a comparison of the baseline condition of the patients prior to LOC with the ROC condition.

In the limitations of the study, it could be questioned whether the doses and stability of a remifentanil perfusion generate equal impacts on the cortical activation effect compared with sufentanil. The beta activity highlighted in the case of inhaled drugs could be due to the use of sufentanil under conditions not comparable with remifentanil (Bolus vs TCI?). Please discuss.

I attach a PDF with my specific comments.

Author Response

Thank you for your constructive feedback. Please see the attachment: our responses to your valuable comments can be found on pages 1 - 4.

Reviewer 2 Report

The article at hand investigates the comparability of EEG-derived cerebral activity measures during emergence from anesthesia maintained with three different regiments. You investigated measures that are recorded using standard monitoring devices, thus the presented data is of direct clinical relevance. The observed EEG markers, derived from alpha and beta frequencies, reveal important differences in power changes during the emergence of responsiveness for the three investigated agents. In connection with differences between the time from EEG changes to actual responses given by patients, that also differed between maintenance anesthetics, the results are of importance not only for anesthesia monitoring during surgery, but also for future development of monitoring devices.

All in all, the research presented is valid and of interest to the community. The methods chosen are adequate and the overall presentation is comprehensive and easy to follow. However, there are some parts that could be improved, from my point of view. I believe the implications of the results for today’s clinical routine could be included a little more within the discussion section. Furthermore, you could elaborate on the underlying neural mechanisms of the anesthetics that might lead to the differences in EEG measures. This would make the manuscripts content more coherent to the title of the special issue. Finally, there are some general suggestions on how to improve the reporting and the presentation of your results.

Please consider the following general suggestions in order to improve your manuscript:

  1. Regarding statistical analysis, please indicate whether you corrected for multiple comparisons and which correction method was used. If no correction was applied, you should discuss this.
  2. Figure 1 should be revised in terms of concordance with the description in the results section. (i) in the text it is referred to figure 1 C-D, however there is no figure 1 D. In addition the description of the colours used in figure 1B got mixed up. Please revise.
  3. Regarding the general presentation of the results, there is some redundant information within the text, figure description, and table 2. Ultimately, it is a matter of preference, however I suggest to report the exact test statistics only once (e.g. within table 2) and otherwise refer to it. Also you should indicate the specific meaning of box-block edges and whiskers.
  4. I recommend to add headings to the subplots displayed in figures (especially figure 2), or to clearly state each subplots’ depiction (e.g. 2B) in order to make it easier for the reader to comprehend all that is shown.

Minor/cosmetic suggestion:

  1. Within the abstract, you could consider only using the LOR/ROR abbreviation in the methods description, as the terms have been introduced earlier. Also, and this is splitting hairs, the introductions of the abbreviations slightly differ (LOR, ROR vs. LOR/ROR).
  2. Page 2, line 3: should read “provide a dimensionless“.
  3. Page 2, line 4: the term level of hypnosis sounds odd in this context.
  4. Page 2, line 24: there is a comma missing behind “care unit [5],“.
  5. Page 2, Section 2.1: paragraph 2: there is a “the“ missing before exclusion criteria.
  6. Page 8, paragraph 2 (Results): there is a mixed up abbreviation. I believe it should be ROR within the second sentence starting: “For PeEN, we observed…“.
  7. Please revise the axis descriptions within Figures 2 and S2, as there are mixed up abbreviations (i.e. time to ROC; and Figure 2 depicts data in relation to ROR).
  8. Page 10, line 8f.: should read “slower oscillatory activity throughout LOR“
  9. Page 10, section 4.1: I don’t quite understand the introduction of the abbreviation REF, as it is not used further, or is there a spelling error?
  10. Page 11, section 4.2: I suggest to combine the first two sentences using a comma, as “both“ refers to the aforementioned frequency bands.
  11. Supplementary Figure S3: Within the figure description there are wrong references to subplots in the final sentences (A, B, C should be changed to B, C, and D).

Author Response

Thank you for your constructive feedback. Please see the attachment: our responses to your valuable comments can be found on pages 5 - 7.
